# Archaeometallurgical Analysis of the Provincial Silver Coinage of Judah: More on the Chaîne Opératoire of the Minting Process

**DOI:** 10.3390/ma16062200

**Published:** 2023-03-09

**Authors:** Maayan Cohen, Dana Ashkenazi, Haim Gitler, Oren Tal

**Affiliations:** 1Department of Archaeology and Ancient Near Eastern Cultures, Tel Aviv University, Ramat Aviv, Tel Aviv 6997801, Israel; 2Leon Recanati Institute for Maritime Studies, University of Haifa, Haifa 3498838, Israel; 3School of Mechanical Engineering, Tel Aviv University, Ramat Aviv, Tel Aviv 6997801, Israel; 4Israel Museum, Derech Rupin 11, Jerusalem 9171002, Israel

**Keywords:** archaeometallurgy, Judah, non-destructive testing analysis, materials characterization, Palestine, silver coins, southern Levant

## Abstract

Silver coins were the first coins to be manufactured by mass production in the southern Levant. An assemblage of tiny provincial silver coins of the local (Judahite standard) and (Attic) *obol*-based denominations from the Persian and Hellenistic period Yehud and dated to the second half of the fourth century BCE were analyzed to determine their material composition. Of the 50 silver coins, 32 are defined as Type 5 (Athena/Owl) of the Persian period Yehud series (ca. 350–333 BCE); 9 are Type 16 (Persian king wearing a jagged crown/Falcon in flight) (ca. 350–333); 3 are Type 24 series (Portrait/Falcon) of the Macedonian period (ca. 333–306 BCE); and 6 are Type 31 (Portrait/Falcon) (ca. 306–302/1 BCE). The coins underwent visual testing, multi-focal light microscope observation, XRF analysis, and SEM-EDS analysis. The metallurgical findings revealed that all the coins from the Type 5, 16, 24, and 31 series are made of high-purity silver with a small percentage of copper. Based on these results, it is suggested that each series was manufactured using a controlled composition of silver–copper alloy. The findings present novel information about the material culture of the southern Levant during the Late Persian period and Macedonian period, as expressed through the production and use of these silver coins.

## 1. Introduction

Analysis of the chemical composition of ancient coins can reveal their alloy content and enable us to determine, among other things, whether certain issues were produced using a controlled and relatively homogenous metallurgical composition. Moreover, the elemental composition of a coin, when compared to a validated dataset, may indicate where that coin was produced [1].

The study of the Provincial silver coinage of Judah was recently revised by Gitler et al. (2023) [2], who presented a typological corpus of the 44 recorded Yehud coin types, as well as a die study of the coins dated to the Late Persian, Macedonian, and Early Hellenistic periods. Here, we present an archaeometallurgical study of some of the more common coin types of the Yehud minting authority (Types 5, 16, 24, and 31) (Figure 1).

Type 5 of the Yehud coinage series (Figure 2) features a helmeted head of Athena in profile, turned to the right on the obverse, while the reverse depicts an owl with the body turned to the right and its head facing front. An olive spray or lily appears in the upper left field, while in the right field, the legend YHD (an abbreviation of the name of the province: Judah) is written in Paleo-Hebrew and Aramaic. These Athenian-styled *gerah* denomination-based issues (1/20 of a local Judahite sheqel; mean weight of 0.48 g based on a sample of 150 coins) were minted in the province of Judah during the Late Persian period between 350 and 333 BCE (see [2]: Chapter III).

With Type 5, the Yehud coinage began a specialization in small denominations in contrast to the larger denominations that had been previously minted for this province (Types 1–4 in the Yehud corpus). Our initial research question was: can we determine with sufficient certainty whether our die-linked issues were not only produced according to a specific standard of metallurgical composition, but also from the same metal batch throughout the minting of the series?

Our study then took this question a step further and sought to determine whether the die-linked issues of Type 5 (Figure 3), which all feature the same obverse (O1) but with five different reverses (R1, R2, R3, R4, and R5), were produced from different metal batches or from the same specific batch that had been used to produce the flans for the entire Type 5 group which is connected to Obverse 1 (O1/R1, O1/R2, O1/R3, O1/R4, and O1/R5 are shown in Figure 2, Figure 3 and Figure 4). Basically, we sought answers that relate to the chaîne opératoire of the earliest small denomination coins minted in the province of Judah.

**Figure 1 materials-16-02200-f001:**
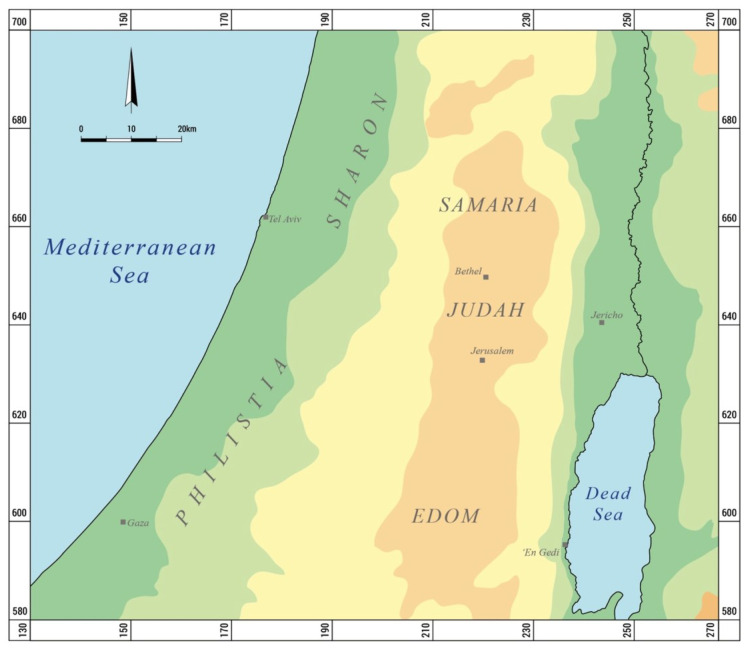
Location map of the research area, showing the fourth-century BCE regional division of southern Palestine.

**Figure 2 materials-16-02200-f002:**
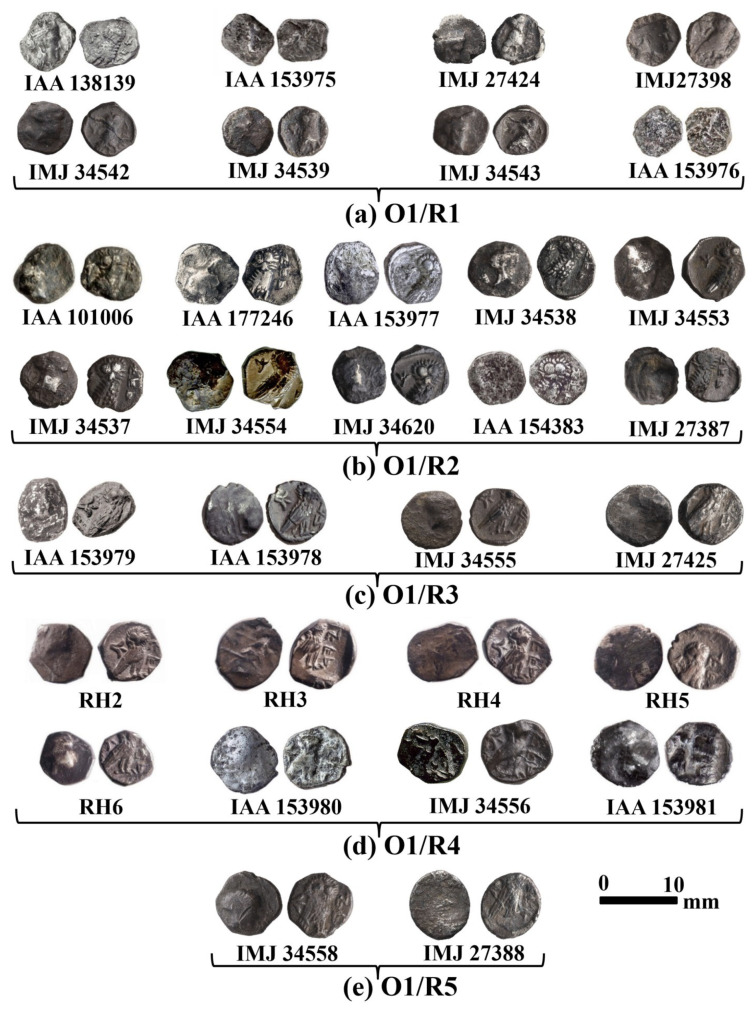
Yehud *gerah* coins—Type 5 of the Yehud series (Athena/Owl) corpus with die combinations O1 (obverse) and R1 (reverse), R2, R3, R4, and R5 ((**a**–**e**), respectively) with the YHD legend.

**Figure 3 materials-16-02200-f003:**
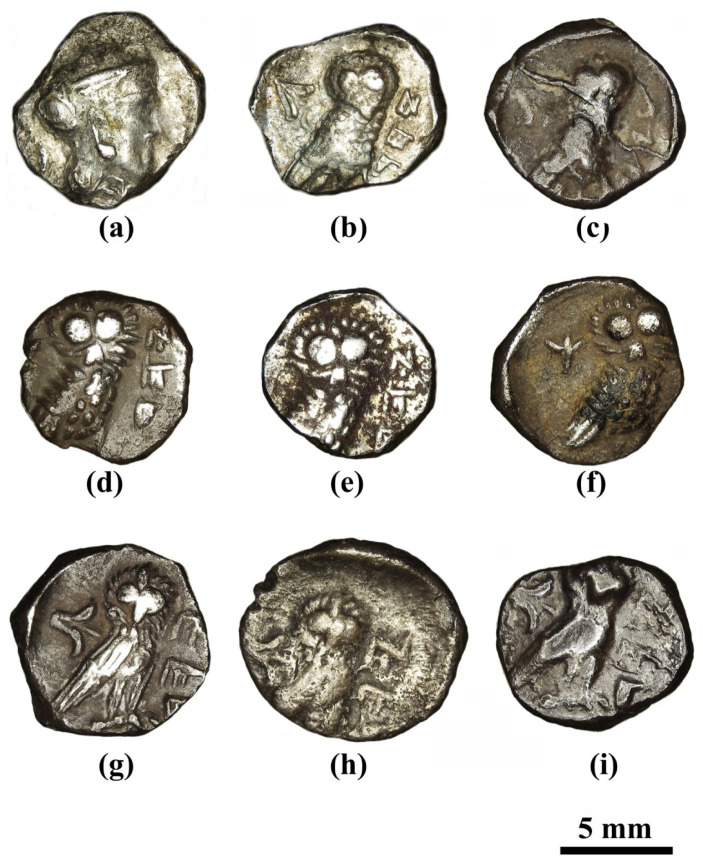
Multi-focal light microscope (LM) enlarged observation of the Yehud *gerah* coins Type 5 die-linked connection, with the same obverse (O1) but different reverses (R1, R2, and R4): (**a**) O1/R1: IAA 138139 (Khirbat Qeiyafa excavations), obverse, showing the helmeted head of Athena; (**b**) O1/R1: IAA 138139, reverse, showing a lily, symbol of Jerusalem (left), owl (center), and Aramaic inscription of Judea (YHD, right); (**c**) O1/R1: IMJ 34542, reverse; (**d**) O1/R2: IMJ 34537, reverse; (**e**) O1/R2: IAA 154383 (Khirbat Qeiyafa); (**f**) O1/R2: IMJ 34538, reverse; (**g**) O1/R4: Ramallah area hoard no. 1, IMJ 2006.53.26139 (RH2), reverse; (**h**) O1/R4: Ramallah area hoard no. 5, IMJ 2006.53.26142 (RH5), reverse; and (**i**) O1/R4: IMJ 34556, reverse.

**Figure 4 materials-16-02200-f004:**
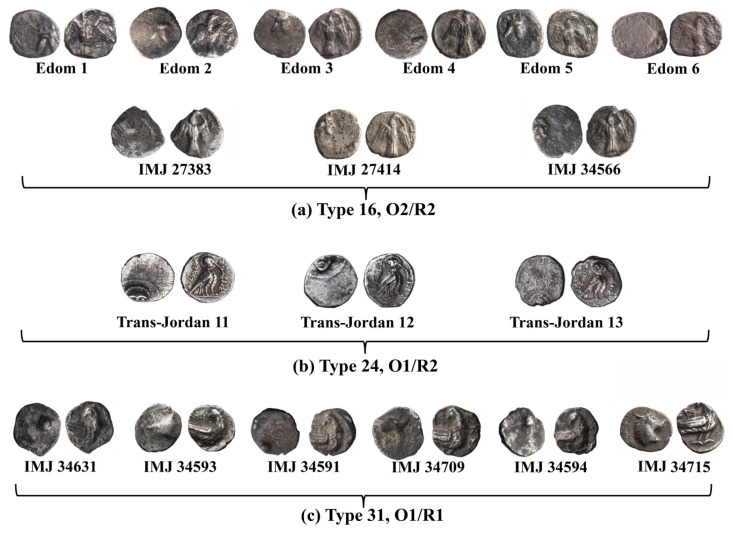
Yehud coins: (**a**) Type 16 O2/R2, (**b**) Type 24 O1/R2, and (**c**) Type 31 O1/R1 of the Yehud series.

Similarly, to other Yehud small denomination coins, Type 5 was struck from dies that were damaged during the striking process but continued to be used (progressive degradation of O1, see [2]: Figure II.1, p. 22). The die damage could have been caused by excessive wear, breaks, or errors. In this peculiar chaîne opératoire, the artisans who struck the coins apparently operated independently of local die engravers and thus were not able to replace the damaged dies or to recut the motifs in order to repair them. Consequently, the damaged obverse dies continued to be used and to produce coins whose obverse motifs are barely discernable [3].

A group of 50 specimens was chosen for this study: 32 coins of Type 5 (Figure 3), 9 coins of Type 16 O2/R2 (Figure 4a), 3 coins of Type 24 O1/R2 (Figure 4b), and 6 coins of Type 31 O1/R1 (Figure 4c). The assemblage includes coins found at controlled archaeological excavations and recorded by the Israel Antiquities Authorities (IAA); coins belonging to the Ramallah area hoard, 2006, which belong to the Israel Museum’s collection [4]; and coins from the Israel Museum’s collection (IMJ).

Because these coins derive from several different sources, we were able to determine whether coins struck from the same pair of dies and found together in the alleged Ramallah area hoard 2006 were produced from the same metal batch or from a different one than that of the coins struck from the same pair of dies and which were either found in controlled archaeological excavations or came from the antiquities market. We assume die-linked coins that appear together in a hoard were probably minted contemporaneously in the striking chain. Hence, our assumption is that their flans were probably prepared from the same metal batch. We also sought to determine the logical striking order of the analyzed Type 5 coins either recovered during controlled archaeological excavations or from the antiquities market based on the deterioration of the obverse and reverse motifs, although such determination is not always precise. We also cannot know how many other coins might have been struck between any two ordered coins in our assemblage during the minting chain process. Our analysis, based on the metallurgical results, may nonetheless help to determine the minting order of a series.

## 2. Technological Background to Research

The study of ancient silver coins includes aspects associated with minting skills, technological abilities, and political and economic considerations of a particular period, and can assist in understanding the connection between political and economic events [5,6,7]. Moreover, it can assist in understanding the economic choices of the minting authorities during times of crisis according to conflicts, as well as during times of reduced supply of metal [8]. For example, throughout history, during periods of high inflation, the silver content of coins is lower than the standard. Hence, the concentration of silver during a specific period can be used as an indicator for the estimation of economic, trade, and social conditions [9].

During antiquity, silver was typically produced from silver-rich galena lead sulfide (PbS) ore containing approximately 1–2 wt% Ag [10]. The cupellation method was a multistage process engaging three separate hearths. The first hearth was used for enriching smelted impure lead that contained silver (bullion). Wood fuel was used in order to remelt the lead bullion at high temperature. The lead was oxidized to litharge (PbO), with a melting temperature of 880 °C, inside a bellows-powered tuyères. Additional bullion was added until an appropriate amount of silver-enriched lead was acquired. Next, the silver-enriched lead was moved to a second hearth and oxidized again; however, at this stage, the litharge was removed by sinking iron rods into the hearth to create coated litharge cones on top of the rods. The rods were then removed, the litharge cones were thrown away, and the rods were dipped again, finally leaving silver globules inside the hearth. In the third hearth, several globules were melted and refined to obtain silver ingots and the remaining lead oxide was absorbed in the porous container wall (cupel). The cupellation method is a very effective process for producing silver metal with more than 95% purity [11].

A partial cupellation process resulted in silver alloy with lead presence. Hence, low quantities or absence of lead in the silver alloy points towards a successful silver refining process [12,13,14]. In addition, high content of lead transforms the silver alloy into a brittle material after a long burial period. Therefore, a good state of preservation of ancient silver objects is often connected to the absence of lead in the alloy.

Pure silver is a shiny white–grey metal with aesthetic appearance; it is a very soft metal that has excellent ductility and malleability [15,16]. Copper was a main alloying element in ancient silver coins. Ancient silver objects, including coins, are commonly available as silver–copper alloys with various ratios of silver and copper [17]. The addition of more than 3 wt% Cu to silver was frequently made to improve the mechanical properties of silver objects and also act as a melting-point depressant. Hence, the presence of more than 3 wt% Cu usually suggests that the copper was intentionally added [12,18,19,20].

Non-invasive and non-destructive testing (NDT) analyses of coins can be rather challenging due to various factors that change the surface metal composition, including long period corrosion processes and the presence of various corrosion products, tarnish and oxide layers, silver enrichment of the surface, cleaning residues, and conservation treatments [19,21,22]. The patina of ancient objects made of silver alloys may contain Ag_2_O, Ag_2_S, and/or AgCl, which can result in uncertainties concerning the obtained elemental data [9]. Therefore, the detected chemical composition obtained from the surface of the coin can be fundamentally different from the chemical composition over the entire volume of the coin [23]. Moreover, even when bulk material composition of ancient silver coins is obtained by destructive testing methods, there may be uncertainties concerning elemental analysis results that assume homogenous elemental distribution, which is often incorrect [21]. Therefore, measuring the composition in several different areas for each coin is recommended [24].

Surface-enrichment can be performed deliberately for technological and/or economic considerations or naturally due to segregation of the metals during casting and cooling stages, and because of corrosion processes [25]. For example, surface-enrichment, making silver–copper coins look like pure silver (silver depletion), was common during the Roman period. When silver–copper coin blanks were cast, they were kept at red heat condition to oxidize the copper on the external surface. Following the copper oxidation, the blanks were dripped into an organic acid bath which removed the copper from the surface, leaving a silver-enriched layer of up to a few hundred microns deep (usually up to a depth of approximately 200 µm). This process was employed by the Romans on silver–copper alloys with a composition of up to 80 wt% Cu, allowing the treated coins to leave the mint looking as if they were made of pure silver [26,27].

Copper located near the external surface of ancient silver coins can be oxidized and form corrosion products after a long burial period in aggressive environments; for example, cuprite (Cu_2_O) and tenorite (CuO) can be formed on the coin surface. When these minerals are removed from the surface, an Ag-rich exterior is achieved depleted in Cu [7]. Nevertheless, ancient silver coins have relatively high durability. In order to examine whether there had been a silver-enrichment of the external surface, Ashkenazi et al. (2017) [19] grounded the surface of a fourth-century BCE silver ring from the Nablus Hoard (Ring B) and found that the Cu wt% concentration at the surface of the ring (before grinding) was similar to the Cu wt% concentration of the bulk alloy (after grinding). This led to the conclusion that the SEM-EDS measurements of the well-preserved fourth-century BCE shiny silver metallic areas of the objects accurately represented the bulk concentration of the silver jewelry from the Samaria and Nablus Hoards. X-ray fluorescence (XRF) and inductively coupled plasma with atomic emission spectrometry (ICP-AES) analyses of southern Palestinian Persian period silver coins also supported this conclusion [19].

## 3. Experimental Methods and Tests

Because of the rareness of the Yehud silver coins, invasive analysis of the objects was not permitted; thus, the coins were studied by using non-destructive testing (NDT) methods:(a)Visual testing (VT) inspection of the coins was performed in order to examine their general preservation condition and locate the better preserved areas of each coin.(b)A handheld X-ray fluorescence (XRF) Oxford X-MET8000 (Oxford Instruments, Abingdon, UK) was employed to examine the obverse and reverse of each coin of group Type 5 O1/R1, O1/R2, and O1/R4 to determine the elemental compositions of the surface. The XRF instrument was combined with a Silicon Drift Detector equipped with a 45 kV Rh Target X-ray tube. Each measurement was performed for 30 s with a detected spot diameter of 5 mm. Oxygen could not be detected with this XRF tool according to instrumental limitation.(c)A multi-focal digital light microscope (LM) (HIROX RH-2000, Hirox, Limonest, France) with high intensity LED lighting (5700 K color temperature) was used to inspect the general preservation of the surface and to detect microscopic discontinuities and defects. The coins were examined with an improved light sensitivity sensor at high resolution HD (1920 × 1200) with a multi-focus system combining many levels of light intensity and an integrated stepping motor, as well as powerful 3D software.(d)Scanning electron microscopy (SEM) with energy dispersive spectroscopy (EDS) examination was performed with environmental SEM in high vacuum mode and an Everhart–Thornley secondary electron (SE) detector. Both secondary electron (SE) and back-scattered electron (BSE) modes were used. The coins’ surface composition (Appendix A) was analyzed by EDS using a Si(Li) liquid-cooled Oxford X-ray detector, calibrated with standard samples from the manufacturer, and providing measurements with a first approximation error of 1% [28]. SEM-EDS analysis accesses only a few micrometers beneath the surface of the analyzed metal; however, it provides very precise information on the morphology of the inspected area [29]. Before SEM-EDS examination, the coins’ surfaces were cleaned with ethanol and dried. In order to measure the coins’ alloy composition, only bright metal regions according to BSE mode were inspected. An example of SEM-EDS spectra of a typical coin’s surface is shown in Appendix A. Different scan areas were examined using EDS between 20 µm × 20 µm and 800 µm × 800 µm. An average composition was then calculated by analyzing each coin using 6–8 measurements from different parts of each coin (both obverse and reverse sides were examined) to collect statistical data on each coin (reliable elemental distribution, average composition values and standard deviation), and then the results were normalized to 100 weight percentage (wt%). The alloy composition was calculated after omitting the peaks of oxides, corrosion products, and soil elements. In order to examine if the surface analysis represented the bulk alloy composition, seven Yehud silver coins were locally ground in different areas with 240–320 silicon carbide grit papers to reveal their bulk metal composition. The examined coins were IAA 153976 and IMJ 27424 (Type 5 O1/R1), IAA 101006 and IAA 154383 (Type 5 O1/R2), IMJ 27383 (Type 16 O2/R2), Trans-Jordan 11 (Type 24 O1/R2), and IMJ 34591 (Type 31 O1/R1). The composition of these coins was detected in several areas before (Appendix A–S8) and after locally grinding their surface (Appendix A).

## 4. Results

VT examination of the external surface of the Yehud *gerah* Type 5 O1/R1, O1/R2, O1/R3, O1/R4, and O1/R5, and of the Yehud series Type 16 (O2/R2), Type 24 (O1/R2), and Type 31 (O1/R1) (Figure 2, Figure 3 and Figure 4) revealed that these coins were well preserved and covered with grey oxide, but also included areas of exposed shiny silver metal. XRF analysis results of the surface of coins Type 5 O1/R1, O1/R2, and O1/R4 revealed high-purity silver with a small percentage (less than 5 wt%) of copper. Other elements were also detected on the surface of the coins, including Si, Au, Pb, Sn, As, Bi, S, Fe, P, Mg, Mn, Ca, Al, and Ti. One coin in the Type 5 group, O1/R1 (IAA 153976), revealed exceptional composition according to XRF analysis with up to 46.7 wt% Ag and 44.5 wt% Cu. Two coins in the Type 5 group, O1/R2 (IAA 154383 and IMJ 27387), also revealed exceptional composition with up to 69.7 wt% Ag and 35.4 wt% Cu (for coin IAA 154383) and with up to 58.4 wt% Ag and 33.1 wt% Cu (for coin IMJ 27387) according to XRF analysis. One coin in group Type 5, O1/R4 (IAA 153981), also revealed exceptional composition according to XRF analysis with up to 76.2 wt% Ag and 12.6 wt% Cu.

### 4.1. Yehud Gerah Type 5 O1/R1

Multi-focal LM observation of the Type 5 O1/R1 coins revealed well-preserved silver metal depicting a helmeted head of Athena on the obverse and an owl on the reverse (Figure 5a–c). On the right field of the reverse of the coins, the Paleo-Hebrew inscription YHD appears (Figure 5a and Appendix A).

The bright areas observed in the SEM BSE mode are silver metal regions and the dark areas according to BSE mode are covered with oxides and some corrosion products (Figure 5d). The SEM-EDS analysis results of eight specimens of the Yehud *gerah* Type 5 O1/R1 coin surfaces (obverse and reverse, Figure 5d, and Appendix A) revealed that the coins were composed of silver, though other elements were also detected, including Cu, Sn, O, Si, Cl, Al, Ca, P, and S (Appendix A).

**Figure 5 materials-16-02200-f005:**
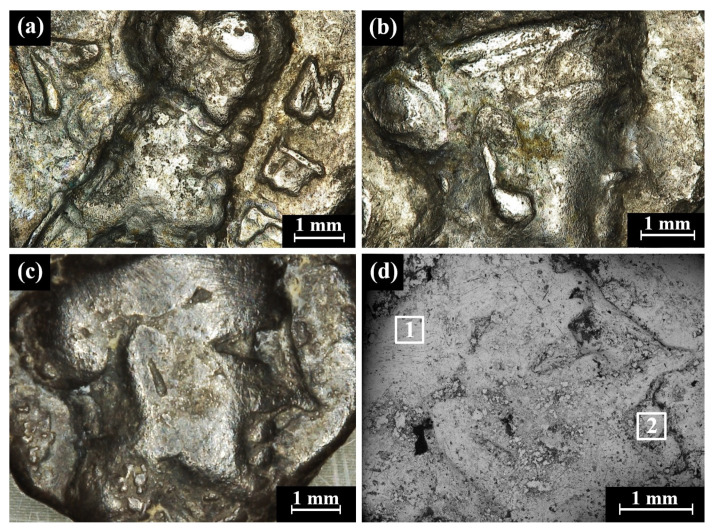
Images of the Yehud *gerah* Type 5 O1/R1 coins: (**a**) IAA 138139 reverse, depicting an owl (multi-focal LM), and (**b**–**d**) obverses of the coin depicting the head of Athena; (**b**) IAA 138139 (multi-focal LM); (**c**) IMJ 27398 (multi-focal LM); and (**d**) IMJ 27398 (SEM, BSE mode), where the white squares (areas 1 and 2) were examined by EDS analysis (the brighter areas are better preserved metal than the darker areas).

The alloy of the coins revealed a composition of 93.2–100 wt% Ag and up to 6.8 wt% Cu (Appendix A), where the average value of the coins’ alloy composition after omitting the peaks of oxides, corrosion products, and soil elements was 98.5 ± 2.0 wt% Ag and 1.5 ± 2.0 wt% Cu (where 42 different areas of the Type 5 O1/R1 coins’ obverse and reverse sides were measured). It seems that a silver content of approximately 97% was the equivalent of “pure silver”. This is entirely to be expected, because elemental silver as measured by modern scientific equipment would not have been available in antiquity. The closest refined silver bullion that could have been achieved by traditional smelting and refining processes would also include traces of gold, lead, and bismuth [2] (p. 334, n. 31).

IAA 153976 coin revealed a different alloy composition of 38.5–100 wt% Ag and up to 61.4 wt% Cu (Appendix A). Therefore, this coin was not included in the average composition value and standard deviation (SD) calculations of group Yehud *gerah* Type 5 O1/R1.

### 4.2. Yehud Gerah Type 5 O1/R2

Multi-focal LM and SEM observations of the Type 5 O1/R2 coins revealed well-preserved silver metal, where the dark areas are covered with oxides and corrosion products and the bright areas represent shiny silver metal (Appendix A).

The SEM-EDS analysis results of 10 specimens of the Yehud *gerah* Type 5 O1/R2 coin surfaces (obverse and reverse) revealed that the coins were composed of silver, though other elements were also detected, including Cu, O, Si, Cl, Al, Ca, Fe, S, Au, and Pb (Appendix A). The bright areas observed in the SEM BSE mode (Figure 6 and Appendix A) are silver metal regions and the dark areas according to BSE mode are covered with oxides and some corrosion products.

The coins’ alloy after omitting the peaks of oxides, corrosion products, and soil elements revealed a composition of 90.2–100 wt% Ag and up to 9.8 wt% Cu, where the average composition value of the alloy of the coins was 96.4 ± 2.5 wt% Ag and 3.6 ± 2.5 wt% Cu (where 53 different areas of the Type 5 O1/R2 coins’ obverse and reverse sides were measured).

IAA 154383 revealed a different alloy composition which is between 42.1–96.9 wt% Ag and 3.1–57.9 wt% Cu. IMJ 27387 also revealed a different alloy composition which is between 29.4–98.0 wt% Ag and 2.0–70.6 wt% Cu (Appendix A). Therefore, these two coins were not included in the average composition value and SD calculations of group Yehud *gerah* Type 5 O1/R2 coins.

### 4.3. Yehud Gerah Type 5 O1/R3 Coins

SEM observation of the reverse of the Yehud *gerah* Type 5 O1/R3, IAA 153978 coin (Figure 7a,b) shows a lily, the symbol of Jerusalem, on the left and an owl on the right side. Only well-preserved silver alloy areas (bright areas according to BSE mode) were examined by EDS analysis (Figure 7b, areas 1–4). SEM-EDS elemental mapping of the reverse of the IAA 153978 coin (Figure 7c,d) revealed that the bright areas, according to the BSE mode, featured silver metal and the dark grey areas were rich in Cl (Figure 7d and Appendix A), while the elements Cu and O (Appendix A, respectively) were distributed relatively homogeneously.

The SEM-EDS analysis results of four specimens of the Yehud *gerah* Type 5 O1/R3 coins (obverse and reverse surfaces) revealed that the coins were composed of silver, though other elements were also detected, including Cu, O, Si, Cl, Al, Ca, S, and Au (Appendix A).

The alloy of the coins after omitting the peaks of oxides, corrosion products, and soil elements revealed a composition of 85.9–100 wt% Ag and up to 14.1 wt% Cu, where the average composition value of the alloy of the coins was 97.6 ± 3.6 wt% Ag and 2.4 ± 3.6 wt% Cu (where 31 different areas of the Type 5 O1/R3 coins’ obverse and reverse sides were measured).

### 4.4. Yehud Gerah Type 5 O1/R4

Multi-focal LM observation of the Ramallah area hoard coins (RH2–RH6) revealed relatively smooth and well-preserved surfaces, with the reverse showing an image of a lily on the left field, an owl, and the Paleo-Hebrew and the inscription YHD (Yeh[u]d, right) (Figure 8a). Yet relatively smooth and well-preserved surfaces, as well as scratches and areas covered with oxides and local corrosion products, were observed by SEM (Figure 8b–d).

SEM-EDS surface analysis results of eight specimens of the Yehud *gerah* Type 5 O1/R4, the reverse of RH2–RH6, IAA 153980, and IMJ 34556 revealed that the coins were composed of silver, though other elements were also detected, including Cu, O, Si, Cl, S, Al, and Ca (Appendix A).

The alloy of seven Yehud *gerah* Type 5 O1/R4 coins revealed a composition of 95.9–100 wt% Ag and up to 4.1 wt% Cu (Appendix A), where the average composition value of the alloy of the coins after omitting the peaks of oxides, corrosion products, and soil elements (O, Si, Cl, S, Al and Ca) was 99.1 ± 1.2 wt% Ag and 0.9 ± 1.2 wt% Cu (where 48 different areas of the Type 5 O1/R4 coins’ obverse and reverse sides were measured).

IAA 153981 revealed a different alloy composition of 81.6–98.0 wt% Ag and 2.0–18.4 wt% Cu (Appendix A). Thus, this coin was not included in the average composition value and SD calculations of group Yehud *gerah* Type 5 O1/R4.

### 4.5. Yehud Gerah Type 5 O1/R5

SEM observations of the Type 5 O1/R5 coins revealed areas of preserved silver metal, with the reverse showing the image of an owl (Figure 9a), where the dark areas according to BSE mode are covered with oxides and corrosion products and the bright areas represent shiny silver metal (Figure 9b). Parallel cracks were observed on the surface of coin IMJ 27388, probably resulting from the striking process and local embrittlement of the silver metal due to local corrosion attack (Figure 9c,d).

The SEM-EDS analysis results of two specimens of the Yehud *gerah* Type 5 O1/R5 coin surfaces (obverse and reverse) revealed that the coins were composed of silver, though other elements were also detected, including Cu, O, Si, Cl, Al, and S (Appendix A).

The alloy of the Yehud *gerah* Type 5 O1/R5 coins after omitting the peaks of oxides, corrosion products, and soil elements revealed a composition of 97.6–100 wt% Ag and up to 2.4 wt% Cu, where the average composition value of the alloy of the coins was 99.7 ± 0.8 wt% Ag and 0.3 ± 0.8 wt% Cu (where 14 different areas of the coins’ obverse and reverse sides were measured).

### 4.6. Yehud Half Gerah Type 16 O2/R2

Multi-focal LM observation of the Edom hoard no. 2 (IMJ 2020.33.2) coin revealed a well-preserved silver metal, with the reverse depicting a falcon in flight (Figure 10a) and the Paleo-Hebrew inscription YHD (Yeh[u]d) on the right field.

The bright areas observed in the SEM BSE mode are silver metal regions and the dark areas according to BSE mode are covered with oxides and some corrosion products (Figure 10d and Appendix A). The SEM-EDS analysis results of nine specimens of the Yehud half *gerah* Type 16 O2/R2 coins (obverse and reverse, Figure 10b–d and Appendix A) revealed that the surface of the coins were composed of silver, though other elements were also detected, including Cu, O, Si, Cl, Al, Ca, Fe, S, Pb, K, and Mg (Appendix A).

The alloy of the Type 16 O2/R2 coins revealed a composition of 98.3–100 wt% Ag and up to 1.7 wt% Cu (Appendix A), where the average composition value of the alloy of the coins after omitting the peaks of oxides, corrosion products, and soil elements was 99.9 ± 0.4 wt% Ag and 0.1 ± 0.4 wt% Cu (where 40 different areas of the Type 16 O2/R2 coins’ obverse and reverse sides were measured).

### 4.7. Yehud Attic Standard Quarter Obol Type 24 O1/R2

VT of the Type 24 O1/R2 Yehud quarter *obol* revealed a preserved silver metal. The obverse depicts a facing head and the reverse an owl standing right, head facing. The bright areas observed in the SEM BSE mode are silver metal regions and the dark areas are covered with oxides and some corrosion products (Figure 11 and Appendix A). The SEM-EDS analysis results of three Yehud quarter *obol* Type 24 O1/R2 specimens (obverse and reverse surfaces, Figure 11 and Appendix A and Appendix A) revealed that the coins were composed of silver, though other elements were also detected, including Cu, O, Si, Cl, Al, Ca, and Fe (Appendix A).

The alloy of the coins revealed a composition of 100 wt% Ag. No presence of Cu was detected in Type 24 O1/R2 coins (and Appendix A, where 20 different areas of the obverse and reverse sides of these coins were measured).

**Figure 11 materials-16-02200-f011:**
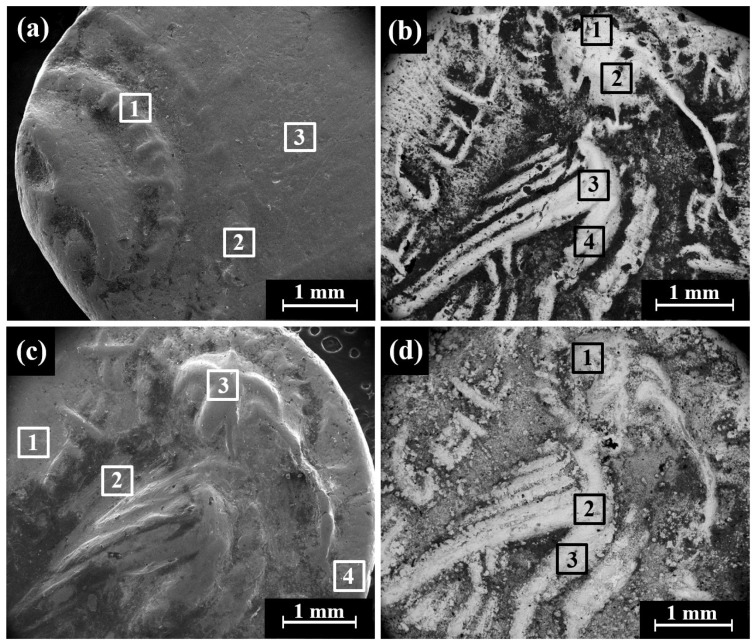
SEM images of the Trans-Jordan hoard coin Type 24 O1/R2: (**a**) coin Trans-Jordan 11 obverse side with a facing head (SE mode); (**b**) coin Trans-Jordan 11 reverse side with and owl (BSE mode); (**c**) coin Trans-Jordan 12 reverse (SE mode); and (**d**) coin Trans-Jordan 12 reverse (BSE mode). The areas 1–4 inside the squares were examined by EDS analysis.

### 4.8. Yehud Attic Standard Hemiobol Type 31 O1/R1

SEM BSE mode observations of the obverse and reverse of Type 31 O1/R1 coins (head of roaring lion/bird standing, right head reverted) revealed bright areas of well-preserved silver metal, though dark areas covered with oxides and corrosion products were also detected (Figure 12 and Appendix A).

The SEM-EDS analysis results of the obverse and reverse of the bright surfaces of six specimens of the Yehud *hemiobol* Type 31 O1/R1 coins revealed that they were mostly composed of silver, though other elements were also detected, including Cu, O, Si, Cl, Al, Ca, Fe, S, and Au (Appendix A). One exceptionally dark surface area of the reverse of coin IMJ 34593 (according to BSE mode) was also detected by EDS (scanned area of 200 µm × 200 µm), showing a composition of 14.1 wt% Cu, 44.8 wt% O, and 35.0 wt% Si, as well as the presence of 2.7 wt% Al, 1.4 wt% Fe, and less than 1.0 wt% of the elements Mg, S, and K. This measurement was not included in the average composition value and SD calculations, since only well-preserved areas were included in the calculations.

The alloy of Type 31 O1/R1 coins after omitting the peaks of oxides, corrosion products, and soil elements revealed a composition of 91.0–100 wt% Ag and up to 9.0 wt% Cu, where the average composition value of the alloy of the coins was 98.3 ± 3.7 wt% Ag and 1.7 ± 3.7 wt% Cu (where 44 different areas of the obverse and reverse sides of Type 31 O1/R1 coins were measured).

**Figure 12 materials-16-02200-f012:**
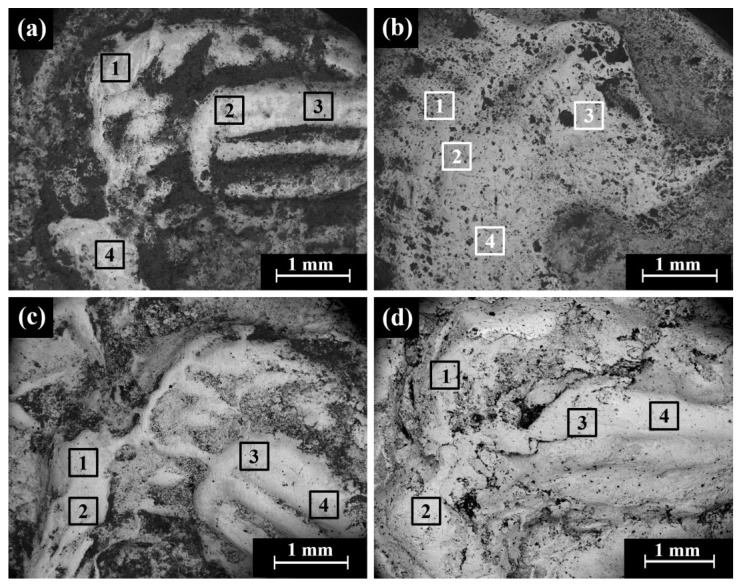
SEM images (BSE mode) of the Yehud *hemiobol* Type 31 O1/R1: (**a**) IMJ 34709 reverse side with an image of a bird standing, right head reverted; (**b**) IMJ 34715 depicts a roaring lion on the obverse; and (**c**) IMJ 34631 and (**d**) IMJ 34594 depict the reverse of these issues. The 1–4 areas inside the squares were examined by EDS analysis.

### 4.9. Chemical Analysis of the Bulk of the Locally Ground Yehud Coins

The exceptional coins IAA 153976 Type 5 O1/R1 and IAA 154383 Type 5 O1/R2, having high copper content (Appendix A, respectively), revealed different copper concentration before and after grinding.

The average copper content in the alloy of coin IAA 153976 before grinding was 27.4 ± 20.7 wt% Cu, while the average copper content in the alloy of coin IAA 154383 alloy before grinding was 22.2 ± 18.9 wt% Cu. Yet the average copper content of the ground areas of the alloy of coin IAA 153976 alloy was 32.7 ± 3.7 wt% Cu and of the ground areas of the alloy of coin IAA 154383 was 16.9 ± 6.6 wt% Cu (Appendix A). The SD of the exceptional coins was reduced dramatically after grinding the surface.

Good agreement was achieved between the surface and bulk compositions of silver coins with low copper content. For example, the average copper concentration of the alloy of coin IMJ 27424 (O1/R1, Appendix A) before grinding was 0.0 ± 0.0 wt% Cu and that of the ground bulk of coin IMJ 27424 was 1.3 ± 1.0 wt% Cu, whereas the average copper content of the alloy of coin IMJ 27383 (Type 16 O2/R2, Appendix A) before grinding was 0.2 ± 0.5 wt% Cu and that of the ground bulk of coin IMJ 27383 was 0.0 ± 0.0 wt% Cu (Appendix A).

## 5. Discussion

The current research, as part of an ongoing study on the early indigenous southern Levantine coinages, employed an archaeometallurgical NDT approach in order to analyze die-linked Late Persian period and Macedonian period Yehud silver coins. In the case of the current coins, based on their shiny metallic appearance, it is evident that the coins have been cleaned. Yet such cleaning procedures were not recorded and remain unknown. Nevertheless, even in such circumstances, surface characterization of silver coins can be performed without any additional polishing of the exterior when the coins’ surfaces are shiny and the remaining oxides and corrosion products layers are thin enough [19,30,31].

All the examined Yehud coins were shown to be in a good state of preservation, based on VT, multi-focal LM, and SEM SE and BSE observations. Such preservation can be explained by the excellent corrosion resistance of silver, due to the formation of stable Ag_2_O on the metal surface [32,33]. Nonetheless, the SEM BSE mode observation combined with EDS analysis revealed the bright areas to constitute exposed silver metal regions and the dark areas as covered with oxides and some corrosion products. The EDS analysis of the surface of the Yehud coins also revealed the presence of the elements O, Si, Cl, Sn, Au, Pb, Al, Ca, Fe, Mg, P, and S (Appendix A). The presence of oxygen and chlorine can arise from both use and storage of the coins and is therefore of little use in shedding light on the metallurgical aspects related to the coins. The absence of lead in most of the examined areas of the coins (lead was only detected by EDS in coin IMJ 34620 (reverse, area 3) of Type 5 O1/R2 and in coin Edom hoard no. 2 (reverse, area 4) of Type 16 O2/R2) indicated that all coins were produced by a successful cupellation refining process. The detection of Cl and S on the surface of the coins was predictable since silver is sensitive to chloride and sulfide ions, leading to the formation of silver chlorides (AgCl) and sulfides (Ag_2_S) as the main contamination products [19,33]. The presence of the elements O, Si, Al, Ca, Fe, Mg, and P is related to the existence of corrosion products and soil remains [6,19,20,33], yet the presence of Fe could also be related to natural impurities in the silver ores [9]. In aggressive environments, silver alloy objects often also contain other corrosion products related to base metals, such as cuprite and litharge [33]. No presence of the elements Cr, Ni, and Cd was detected. These elements are typical of modern forgeries [24], indicating the authenticity of all examined silver coins.

The presence of gold in silver alloys was apparently unfamiliar to the ancient minters. Yet ancient silver coins often contain less than 1 wt% Au as an impurity associated with the silver ore. Therefore, the gold content in silver coins can assist to distinguish between silver sources used for coinage [34].

Good agreement was observed between the SEM-EDS analysis results of the surface and the bulk after grinding the surface, and between the SEM-EDS analysis results and the XRF analysis results of Type 5 O1/R1, O1/R2, and O1/R4 coins’ surfaces, where with both methods, high-purity silver was detected, with an average copper content of less than 5 wt% Cu, as well as other elements, including, O, Si, Sn, Au, Pb, Al, Ca, Fe, Mg, P, and S. The elements Mn, As, Bi, and Ti were also detected by XRF analysis on the coins’ surface but were not detected by EDS analysis, whereas the presence of Cl was detected in the EDS analysis but not by the XRF analysis.

Because of the importance of silver as a representative of the material cultural heritage of different populations throughout history, numerous studies of ancient silver objects, including coins, exist in the literature, studying the chemical composition, microstructure, provenance, manufacturing processes, and the corrosion products of silver items. These studies usually combine NDT and destructive testing methods, for example, LM and SEM observation of metallographic samples, particle-induced X-ray emission (PIXE) analysis, inductively coupled plasma (ICP) analysis, X-ray diffraction (XRD), and X-ray photoelectron spectroscopy (XPS) analysis [19,22,23,27,28]. Some possible NDT methods that can be applied for elemental bulk analysis of silver coins are the neutron activation analysis (NAA) and the prompt-gamma activation analysis (PGAA) [34,35,36,37,38,39]. Yet these methods have some limitations resulting from the high cross section for neutron capture by silver itself, and therefore the silver data would prevent revealing trace elements that could be useful fingerprint variations in the minting process. There are certain rare-earth metallic elements with large neutron cross sections that could provide interesting data concerning differences in the raw silver source. Another NDT technique that could be used to provide additional information concerning the coins’ bulk material composition is the muon induced X-ray emission (MIXE) technique [40].

Ancient silver was commonly produced from silver-rich galena lead sulfide ore using the cupellation process [10]. The presence of Pb and S in some of the measurements is thus probably related to the cupellation process [11,21]. The cupellation technique is very effective for the production of high-purity silver metal, with more than 95 wt% Ag [11], which may explain the high purity of the discussed Yehud silver coins. While the presence of Au and Sn might be related to the ore deposits that were used, the presence of Sn might also be related to the addition of copper to the alloy.

Copper was a main alloying element in ancient silver coins and the addition of copper to silver was often performed to depress the melting point as well as to improve the mechanical properties of the alloy [12,18,19,20,22]. Since the presence of Sn is quite rare in galena ores, its presence might point to alloying the silver with bronze instead of pure Cu [21]. According to Brocchieri et al. (2020), the detection of high-purity silver coins probably indicates that during the production process of these coins, the mint did not suffer from economic constraints [5]. In addition to the Ag and Cu alloy elements and the corrosion products and soil elements that were detected in all Yehud series coins, other elements were also detected. For example, up to 4.7 wt% Sn was detected in a specific 300 μm × 300 μm scanned area of IAA 153975 (Type 5 O1/R1, Appendix A). Up to 1.1 wt% Au was detected in the reverse of coin IMJ 34553 and up to 1.2 wt% Au was detected in the obverse of this coin (Type 5 O1/R2, Appendix A). In addition, up to 1.7 wt% Pb was detected in a specific 300 μm × 300 μm scanned area of IMJ 34620 reverse (Type 5 O1/R2, Appendix A). This may also support the assumption that each series was manufactured by using a specific composition of silver–copper alloy.

Although SEM-EDS is a valuable NDT tool for surface analysis of ancient silver coins, if the detected objects are covered with a thick oxide layer and contain massive corrosion products, the analysis may not provide a representative characterization of the bulk alloy composition of the object [19,30,41]. Ancient silver objects such as coins sometimes exhibit silver enrichment of the surface and, as a result, a considerable reduction of copper on the surface [33,42]. In silver objects retrieved from a soil environment after a long burial period, a local selective galvanic corrosion attack on the copper-enriched areas may occur, resulting in the diffusion of Cu from the bulk of the Ag object to its surface, causing the formation of a cuprite layer on the external surface of the coin [43,44]. Moreover, under some circumstances, unsuitable conservation methods may cause damage to the ancient coins and affect their surface composition. Nevertheless, when the oxide layer is thin and the shiny metal is exposed beneath, the limitations of SEM-EDS analysis should not prevent the use of this method as a valuable tool for the characterization of ancient silver coins. However, it is essential first to determine whether the composition of the surface layer is representative of the bulk composition of the object [18,19,32,42,45]. In order to do so, seven Yehud silver coins were locally ground in several areas to expose their bulk metal and their composition was examined before and after grinding. Good agreement was achieved between surface and bulk compositions of coins with low copper content (Appendix A).

In order to obtain reliable results from the EDS surface analysis, only bright areas with a shiny silver metal appearance were examined here for the calculations of the average value and SD of the alloy composition of each group of coins (Table 1). Our results, after eliminating the peaks of oxides and soil elements, revealed that five of the die combination issues of the Yehud series Type 5 (O1: R1, R2, R3, R4, R5), Type 16 (O2/R2), Type 24 (O1/R2), and Type 31 (O1/R1) are composed of high-purity silver with a small percentage of copper (Table 1).

The copper content in the Type 5 coin alloy is between 0.3 ± 0.8 wt% Cu (for O1/R5) and 3.6 ± 2.5 (for O1/R2). The copper content in the Type 16 coin alloy is 0.1 ± 0.4 wt% Cu (for O2/R2); the copper content in the Type 24 O1/R2 is 0 ± 0 wt% Cu; and the copper content in the Type 31 O1/R1 is 1.7 ± 3.7 wt% Cu (Table 1). Four coins (IAA 153976 of Type 5 O1/R1, IAA 154383 and IMJ 27387 of Type 5 O1/R2, and IAA 153981 of Type 5 O1/R4) were not included in the average composition values and SD calculations of the main group of each coins. The fact that these four coins contain a high amount of copper (20.7 ± 19.3 wt% Cu, Table 1) may indicate that towards the end of the minting process of each series there was a shortage of raw materials and therefore recycled silver was used.

The manufacturing processes of the coins from all the examined groups were similar (casting and striking a blank flan), with some slight differences (between pure silver alloy for the case of Type 24 O1/R2 coins up to 3.6 ± 2.5 wt% Cu for the case of Type 5 O1/R2 coins). Three additional die-connected Yehud coins of the late addition type (a female {?} head to the right with the Aramaic letter yod in the left field on the obverse and an owl standing right, head facing on the reverse) were studied previously (Table 1) [2] and are used here as a reference group with an average composition of 0.4 ± 0.8 wt% Cu, presenting the similarities and differences in terms of metallurgical composition compared with the above-mentioned coin types.

For comparison, the alloy composition of 80 selected Persian period Samarian silver coins from the Samaria and Nablus Hoards, as well as other coins from the Israel Museum collection, was 95.9 ± 2.5 wt% Ag and 4.1 ± 2.5 wt% Cu (SEM-EDS analysis after omitting the peaks of oxides, corrosion products, and soil elements) [19]. Moreover, the average alloy composition of the Persian period jewelry from the Samaria Hoard was 93.4 ± 1.65 wt% Ag and 6.6 ± 1.6 wt% Cu, while that of the jewelry from the Nablus Hoard was 94.9 ± 1.9 wt% Ag and 5.1 ± 1.9 wt% Cu (excluding the joining areas). In order to determine with sufficient certainty whether the current studied die-linked silver coins were produced from the same metal batch throughout the minting of each group, the copper concentration distribution of each series is shown based on the SEM-EDS analysis results after omitting the peaks of oxides, corrosion products, and soil elements (Appendix A), presenting the Cu wt% concentration range vs. the relative no. of measurements (%). Based on the average values and SD of the alloy composition of each series and the copper concentration distribution of the different groups, each series (including the die-linked issues of Type 5 O1/R1, O1/R2, O1/R3, O1/R4, and O1/R5 subtypes, as well as Type 16 O2/R2, Type 24 O1/R2, and Type 31 O1/R1) was most probably manufactured by using a controlled specific composition of silver–copper alloy.

Based on the current findings, a four-step methodology is suggested for the study of ancient silver coins: (first) a VT should be performed on the objects; (second) the coins should be examined by multi-focal LM in order to establish the areas where shiny silver metal is exposed and whether there are corrosion products on the surface of the coin (the silver alloy composition should be measured only in these shiny areas); (third) initial examination of the coins’ surface should be conducted by XRF; and (fourth) the coins should be examined by SEM-EDS analysis in both SE and BSE modes in order to determine the state of preservation of each coin and its alloy composition. Only bright areas observed in the BSE mode should be subjected to EDS analysis in order to determine the alloy composition of each series of coins.

The elaborate iconographic designs on these tiny Persian period and Macedonian period Yehud silver coins demonstrate high artistic and technological abilities (Appendix A). Based on the average alloy composition values and SD of the examined series of the five die combination issues of the Yehud series Type 5 (O1 linked with R1, R2, R3, R4, and R5), each series was struck from a different and controlled specific composition of silver–copper alloy. Approximately 10% of the coins revealed a different alloy composition, however, with a much higher amount of copper and a heterogeneous composition. This implies that at a certain stage of the minting process, a different batch of, possibly recycled, alloy was used rather than the standardized alloy that was recorded for all the other coins of the same die connection.

## 6. Conclusions

In this study, 50 Late Persian period and Macedonian period silver coins were studied by NDT metallurgical analysis. The results obtained from the SEM-EDS surface analysis of well-preserved areas sufficiently represent the concentrations of the coins’ ground bulk. Correspondence was received between the SEM-EDS analysis results and the XRF analysis results. The results show that the coins are made of high purity silver with only small percentages of copper. All the coins were produced by similar techniques of casting flans and striking them by plastic deformation. Based on the average alloy composition values and SD, as well as the copper concentration distribution of the examined series, the five die combination issues of the Yehud series Type 5 O1/R1–O1/R5, Type 16 O2/R2, Type 24 O1/R2, and Type 31 O1/R1 groups, and the late addition Yehud coin type, each series was most likely produced by a different and controlled specific composition of silver–copper alloy. However, four Type 5 exceptional coins (IAA 153976 O1/R1, IAA 154383 O1/R2, IMJ 27387 O1/R2, and IAA 153981 O1/R4) revealed a different alloy composition with a much higher amount of copper and heterogeneity. This implies that at a certain stage of the minting process, a different batch of possibly recycled alloy was used instead of the standardized alloy that was recorded for all other coins of the same die connection. The current four-step methodology revealed novel information concerning the material culture of the southern Levant during the Late Persian period associated with minting production of silver coins. This four-step methodology can be used with additional bulk NDT methods by other researchers to study various ancient silver objects.

## Figures and Tables

**Figure 6 materials-16-02200-f006:**
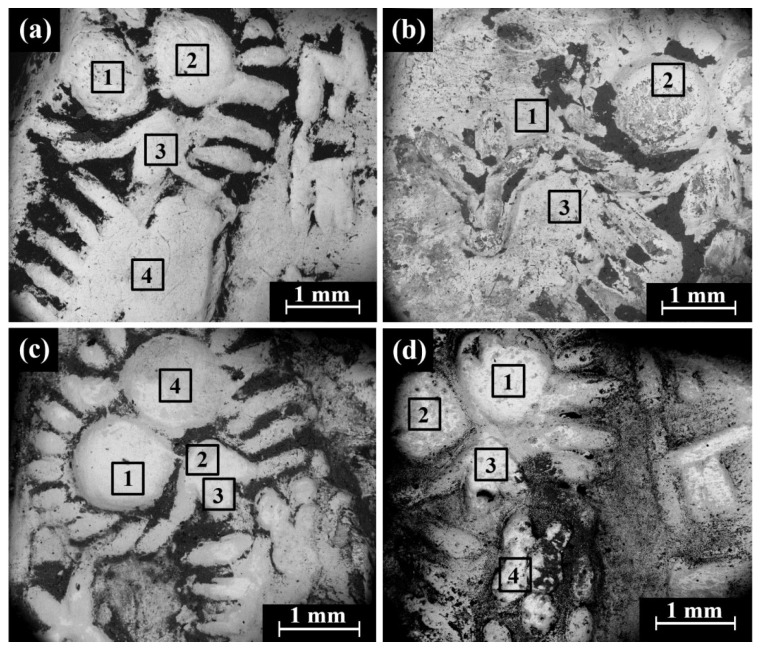
SEM images of the Yehud *gerah* Type 5 O1/R2 coins, with the reverse depicting an owl (BSE mode): (**a**) IAA 177246; (**b**) IAA 153977; (**c**) IMJ 34553; and (**d**) IMJ 34537. The bright areas according to BSE mode (areas 1–4 inside the squares) were examined by SEM-EDS analysis.

**Figure 7 materials-16-02200-f007:**
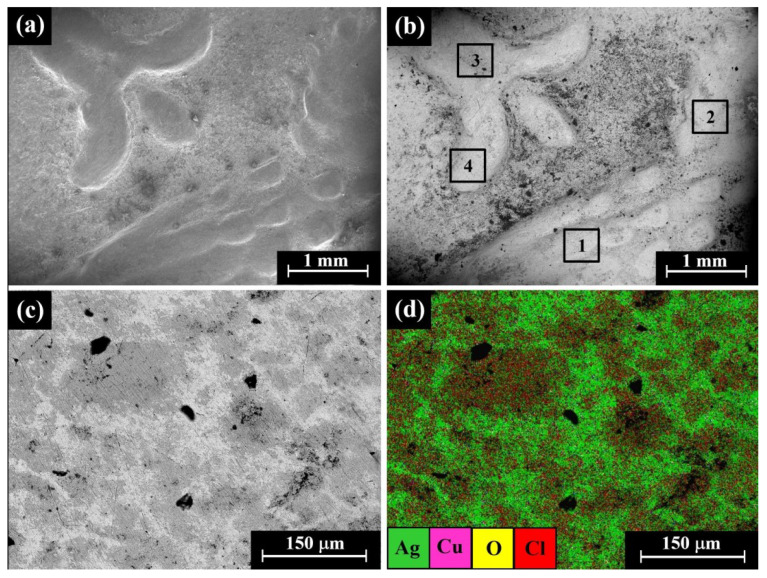
SEM images of the Yehud *gerah* Type 5 O1/R3 coin IAA 153978: (**a**) reverse, images of a lily and an owl, SE mode; (**b**) reverse, BSE mode, where the bright areas 1–4 inside the squares were examined by EDS analysis; (**c**) obverse, BSE mode; and (**d**) elemental mapping, where the green areas are rich in silver and red areas are rich in chlorine.

**Figure 8 materials-16-02200-f008:**
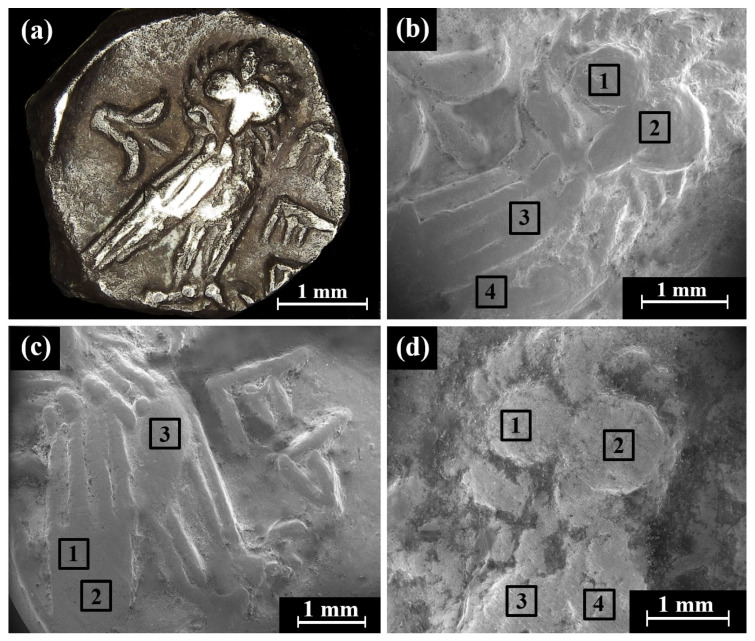
Images of the Yehud *gerah* Type 5 O1/R4, the reverse of the Ramallah area hoard coins (RH): (**a**) general view of coin RH2 (multi-focal LM); (**b**) RH2 (SEM, SE mode); (**c**) RH3 (SEM, SE mode); and (**d**) RH5 (SEM, SE mode). The areas 1–4 inside the squares were examined by SEM-EDS analysis.

**Figure 9 materials-16-02200-f009:**
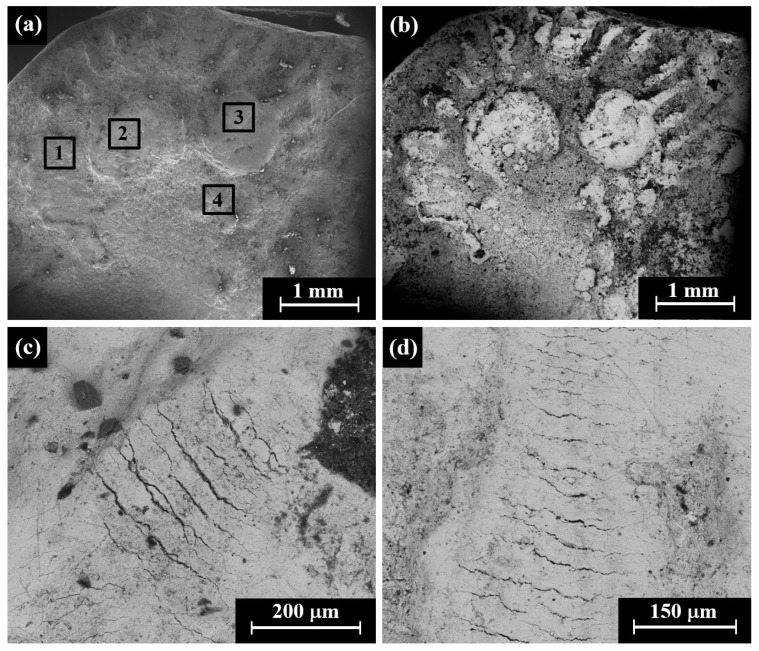
SEM images of the Yehud *gerah* Type 5 O1/R5 reverse side with the image of an owl: (**a**) coin IMJ 34558 (SE mode), where the areas inside the squares 1–4 were examined by EDS analysis; (**b**) coin IMJ 34558 (BSE mode); and (**c**,**d**) coin IMJ 27388 with parallel cracks.

**Figure 10 materials-16-02200-f010:**
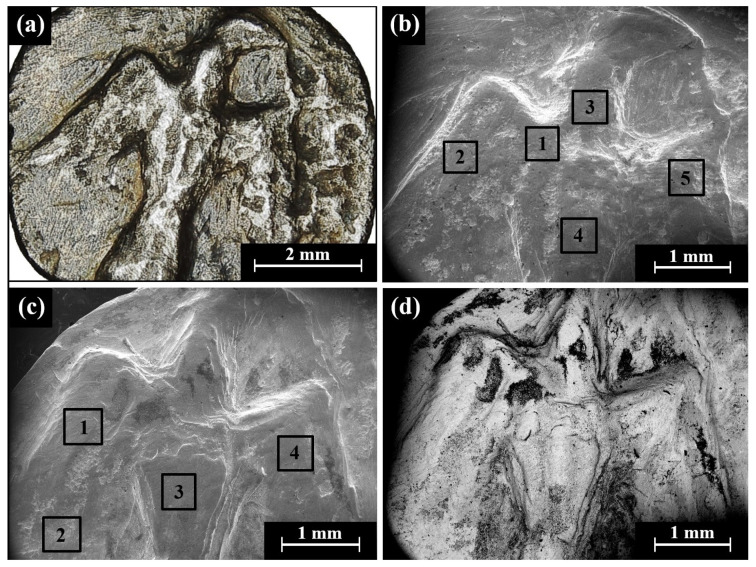
Images of the Edom hoard Type 16 O2/R2 coins, with the reverse side showing a design of a falcon in flight: (**a**) multi-focal LM of coin Edom hoard no. 2 (IMJ 2020.33.2); (**b**) SEM observation of coin Edom hoard no. 2 (SE mode); (**c**) coin Edom hoard no. 3 (SE mode); and (**d**) coin Edom hoard no. 3 (BSE mode). The areas inside the squares 1–5 in (**b**,**c**) were examined by EDS analysis.

**Table 1 materials-16-02200-t001:** The average alloy composition (Cu wt% content) of the different Yehud Types 5, 16, 24, and 31, the late addition Yehud coins, and the Samaria and Nablus Hoards coins, according to SEM-EDS analysis.

Coin Types	No. of Examined Items	No. of Tests Performed by SEM-EDS	Average Cu (wt%) Content in the Ag-Cu Alloy
Yehud *gerah* Type 5 O1/R1 coins	7	42	1.5 ± 2.0
Yehud *gerah* Type 5 O1/R2 coins	8	53	3.6 ± 2.5
Yehud *gerah* Type 5 O1/R3 coins	4	31	2.4 ± 3.6
Yehud *gerah* Type 5 O1/R4 coins	7	48	0.9 ± 1.2
Yehud *gerah* Type 5 O1/R5 coins	2	14	0.3 ± 0.8
Yehud *gerah* Type 5 coins with exceptional composition	4	28	20.7 ± 19.3
Yehud half *gerah* Type 16 O2/R2 coins	9	40	0.1 ± 0.4
Yehud quarter *obol* Type 24 O1/R2 coins	3	20	0.0 ± 0.0
Yehud *hemiobol* Type 31 O1/R1 coins	6	44	1.7 ± 3.7
Late addition to the Yehud corpus [2]: 270–271, Figure VIII, 84–85	3	22	0.4 ± 0.8
Coins from the Samaria and Nablus Hoards [19]	80	160	4.1 ± 2.4

## Data Availability

The data supporting the reported results can be found in the Appendix A file.

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
