# Peer review of "Archaeometallurgical Analysis of the Provincial Silver Coinage of Judah: More on the Chaîne Opératoire of the Minting Process"

_materials, 2023, doi:10.3390/ma16062200_

Round 1

Reviewer 1 Report

The paper reports on the comprehensive study of 50 late Persian period silver coins using a four-step non-destructive testing methodology. This methodology allowed to determine of chemical composition of the coins and to make a supposition about the material culture associated with minting production of silver coins in the Southern Levant.

The subject of the paper fits the scope of Materials journal.

The paper is suitable for publication in its present form.

additional comments:

1. The methodology proposed by the authors allowed for non-destructive testing of rare silver coins. This methodology could further be used in the subsequent studies of other rare and unique silver objects, namely ancient jewelry, coins, etc.

2. The quality of the figures is good enough while the photographs of the coins in the manuscript should be improved. 

Author Response

Reviewer 1: Additional comments:

1. The methodology proposed by the authors allowed for non-destructive testing of rare silver coins. This methodology could further be used in the subsequent studies of other rare and unique silver objects, namely ancient jewelry, coins, etc.

Answer: We thank the reviewer for his positive opinion. The English language spell mistakes in the manuscript were corrected. More relevant references were added to the manuscript.

Changes in the text: 

Introduction, page 4, line 92.

Technological Background to Research, page 6, line 150.

Experimental Methods and Tests, page 7, lines 224-229.

Discussion, page 18, line 570.

References, page 23, lines 812-825.

2. The quality of the figures is good enough while the photographs of the coins in the manuscript should be improved.

Answer: The photographs of the coins were improved and high resolution photographs of the coins were added to the Electronic Supplementary Material file (Figs. S9-S16).

For more information please see the attached file.

Reviewer 2 Report

The manuscript is generally well written and clear.  However, I think that it can be improved in certain places.  My brief comments follow:

1.  On p. 4, line 90, "A group of 50 specimen were..." should be stated "A group of 50 specimens was...".

2.  On p. 6, line 149, I think that "various ratios silver" should be "various ratios of silver". 

3. On p. 7, line 222, I think the authors mean "average composition was then calculated...", not than.  And below on line 224,  I don't understand the term "to collect a statistically calculated information". 

4.  The authors use the term "ratio" often but rarely give actual ratios.  For instance, on p. 15, lines 442-445, the authors start with "The Ag/Cu ratio" but then only give measured weight percents of Ag and Cu, never a ratio.  It is a minor point but one that begs consistency.

Author Response

Reviewer 2:

Answer: We thank the reviewer for his positive opinion and comments. The English mistakes were corrected. Relevant references were added and the manuscript was improved correspondingly.

Comments and Suggestions for Authors:

The manuscript is generally well written and clear.  However, I think that it can be improved in certain places.  My brief comments follow:

1. On p. 4, line 90, "A group of 50 specimen were..." should be stated "A group of 50 specimens was...".

Answer: The text was corrected.

Change in the text:

Introduction, page 4, line 92.

2. On p. 6, line 149, I think that "various ratios silver" should be "various ratios of silver". 

Answer: The text was corrected.

Change in the text:

Technological Background to Research, page 6, line 151.

3. On p. 7, line 222, I think the authors mean "average composition was thencalculated...", not than.  And below on line 224,  I don't understand the term "to collect a statistically calculated information". 

Answer: The text was corrected.

Change in the text:

Experimental Methods and Tests, page 7, lines 224-229.

3. The authors use the term "ratio" often but rarely give actual ratios.  For instance, on p. 15, lines 442-445, the authors start with "The Ag/Cu ratio" but then only give measured weight percepts of Ag and Cu, never a ratio.  It is a minor point but one that begs consistency.

Answer: We thank the reviewer for this constructive comment. We removed the term “The Ag/Cu ratio” from the manuscript and changed the text correspondingly.

Changes in the text:

Results, page 9, lines: 273, 298.

Results, page 11, line 332.

Results, page 12, line 353.

Results, page 13, line 379.

Results, page 14, line 403.

Results, page 15, lines 419, 444.

For more information please see the attched file.

Reviewer 3 Report

Excellent research. Please supplement the spectra from the results of the SEM-EDS analysis.

Author Response

Reviewer 3:

Answer: We thank the reviewer for the positive opinion concerning our manuscript. The English language spell mistakes were corrected.

Changes in the text:

Introduction, page 4, line 92.

Technological Background to Research, page 6, line 150.

Experimental Methods and Tests, Page 7, lines 224-229.

Results, page 16, line 465.

Discussion, page 18, line 571.

Comments and Suggestions for Authors: Excellent research. Please supplement the spectra from the results of the SEM-EDS analysis.

Answer: An example of SEM-EDS spectra of a typical coin’s surface was added to the Electronic Supplementary Material.

Change in the text:

Experimental Methods and Tests, page 7, lines 222-223.

Electronic Supplementary Material, page 1, Fig. S1.

For more information please see the attached file.

Reviewer 4 Report

Review of  Archaeometallurgical Analysis of the Provincial Silver Coinage of Judah: More on the C_h_a_în_e_ _O_p_ér_a_t_o_i_r_e_ _of the Minting Process

I have reviewed this paper and the supplemental material as well.  The introduction lays out an interesting problem, the body of the paper contains a useful set of data from non-destructive analytical methods.   However, the conclusions from these data are weak.   Surely, there is more both chemical and archeological information that can be gleaned from this study.   

I wondered if the authors had given thought to other non-destructive methods of analysis for elements in the bulk of the coins.  The data for both oxygen and chlorine are of little use in shedding light on the metallurgy as they can arise in both use and storage.  Are the variations in aluminum content of interest, or is it also a post-production contaminant?  Gold and tin are another matter, however

One possibility for bulk analysis might be both instrumental neutron analysis and prompt-gamma analysis.   The disadvantage of those methods is the high cross section for neutron capture by silver, itself, to the point that the silver data would prevent finding much useful trace-elements that could fingerprint variations in the minting process.  There are some rare-earth elements with large neutron cross sections that could provide interesting information about variations in the source of the raw silver stock.

Another method that extends analysis deeper into the coin would be muon-x-ray analysis MIXE.  There is a facility at PSI in Zurich for such analyses.  Although it is comparable to PIXIE, it is possible to probe deeper into the coin, and determine elemental depth analyses.

In view of the likelihood that these data will surely be of value to others working in this field, I would recommend publication of this work after the authors have provided some more interesting conclusions from their work.  

Author Response

Reviewer 4:

Answer: The English language spell mistakes in the manuscript were corrected. The discussion and conclusions parts were improved based on the reviewer’s remark and seven new references were added to the manuscript correspondingly. The information about the corrections and additions we made to the manuscript is presented below.

Comments and Suggestions for Authors:

I have reviewed this paper and the supplemental material as well.  The introduction lays out an interesting problem, the body of the paper contains a useful set of data from non-destructive analytical methods.  However, the conclusions from these data are weak.  Surely, there is more both chemical and archeological information that can be gleaned from this study.  

Answer: We thanks the reviewer for his important and constructive comments. We improved the manuscript correspondingly.

I wondered if the authors had given thought to other non-destructive methods of analysis for elements in the bulk of the coins.  The data for both oxygen and chlorine are of little use in shedding light on the metallurgy as they can arise in both use and storage.  Are the variations in aluminum content of interest, or is it also a post-production contaminant?  Gold and tin are another matter, however.

One possibility for bulk analysis might be both instrumental neutron analysis and prompt-gamma analysis.  The disadvantage of those methods is the high cross section for neutron capture by silver, itself, to the point that the silver data would prevent finding much useful trace-elements that could fingerprint variations in the minting process.  There are some rare-earth elements with large neutron cross sections that could provide interesting information about variations in the source of the raw silver stock.

Another method that extends analysis deeper into the coin would be muon-x-ray analysis MIXE.  There is a facility at PSI in Zurich for such analyses.  Although it is comparable to PIXIE, it is possible to probe deeper into the coin, and determine elemental depth analyses.

In view of the likelihood that these data will surely be of value to others working in this field, I would recommend publication of this work after the authors have provided some more interesting conclusions from their work. 

Answer: The discussion and conclusions parts were improved based on the comments of the reviewer. In addition seven references were added to the manuscript.

Changes in the text:

Introduction, page 4, line 92.

Technological Background to Research, page 6, line 150.

Experimental Methods and Tests, page 7, lines 222-223.

Experimental Methods and Tests, Page 7, lines 224-229.

Results, page 9, line 273, lines 298-299.

Results, page 16, line 465.

Discussion, page 17, lines 492-493.

Discussion, pages 17-18, lines 506-534.

Conclusions, page 20, lines 644-663.

References, page 23, lines 812-825.

For more information please see the attaced file.
